# Effects of Ground Subsidence on Vegetation Chlorophyll Content in Semi-Arid Mining Area: From Leaf Scale to Canopy Scale

**DOI:** 10.3390/ijerph20010493

**Published:** 2022-12-28

**Authors:** Xingchen Yang, Shaogang Lei, Yunxi Shi, Weizhong Wang

**Affiliations:** 1Engineering Research Center of Ministry of Education for Mine Ecological Restoration, China University of Mining and Technology, Xuzhou 221116, China; 2School of Environment and Spatial Informatics, China University of Mining and Technology, Xuzhou 221116, China; 3Environmental Restoration and Management Center of Jungar Banner Mining Area, Ordos 017100, China

**Keywords:** coal mining, ground subsidence, chlorophyll content, leaf scale, canopy scale

## Abstract

Ground subsidence is the main cause of vegetation degradation in mining areas. It is of great significance to study the effects of ground subsidence on vegetation. At present, few studies have analyzed the effects of ground subsidence on vegetation from different scales. However, the conclusions on different scales may differ. In this experiment, chlorophyll content was used as an indicator of vegetation degradation. We conducted a long-term field survey in the Lijiahao coalfield in China. Based on field survey data and remote sensing images, we analyzed the effects of ground subsidence on chlorophyll content from two scales (leaf scale and canopy scale) and summarized the similarities and differences. We found that, regardless of leaf scale or canopy scale, the effects of subsidence on chlorophyll content have the following three characteristics: (1) mining had the least effect on chlorophyll content in the neutral area, followed by the compression area, and the greatest effect on chlorophyll content in the extension area; (2) subsidence had a slight effect on chlorophyll content of *Caragana korshins*, but a serious effect on chlorophyll content of *Stipa baicalensis*; (3) chlorophyll content was not immediately affected when the ground sank. It was the cumulative subsidence that affects chlorophyll content. The difference between leaf scale and canopy scale was that the chlorophyll content at canopy scale is more affected by mining. This means that when assessing vegetation degradation, the results obtained by remote sensing were more severe than those measured in the field. We believe that this is because the canopy chlorophyll content obtained by remote sensing is also affected by the plant canopy structure. We recommend that mining and ecological restoration should be carried out concurrently, and that ground fissures should be taken as the focus of ecological restoration. In addition, *Caragana korshins* ought to be widely planted. Most importantly, managers should assess the effects of ground subsidence on vegetation on different scales. However, managers need to be aware of differences at different scales.

## 1. Introduction

Coal mining is very important to China’s economic development. At present, coal is still the cornerstone of China’s energy security, accounting for about 57% of its primary energy. Wang pointed out that coal will still account for more than half of China’s primary energy by 2030 [1]. Therefore, in the short term, large-scale coal mining will continue in China.

However, coal mining will cause ecological degradation, especially vegetation degradation. Coal resources in the west of China account for more than 70% of the total coal resources [2]. Unfortunately, western China has been defined as an ecologically fragile zone because of the arid soils and low rainfall climate that render ecosystem restoration problematic [3]. Existing studies have shown that coal mining has caused serious damage to the ecological environment in the semi-arid region, mainly including groundwater level decline [4,5], soil moisture reduction [6,7], soil structure deterioration, and vegetation degradation [8,9,10,11]. Among these, vegetation, as the most intuitive representation of environmental degradation, has always been the focus of ecological restoration in mining areas. Vegetation degradation can be reflected in many aspects, such as vegetation coverage [12], biomass [13], and chlorophyll content [14]. Chlorophyll content is a valuable diagnostic indicator for the early identification and assessment of the overall health of vegetation, indicating its degradation status [15,16]. Many studies have shown that chlorophyll content is an important factor that should be examined to determine plant stress conditions [17,18].

Underground mining causes ground subsidence [19,20], which is the main cause of vegetation degradation [14]. Ma et al. pointed out that subsidence caused by mining activities had a certain impact on the surface vegetation in the mining areas [21]. Wang et al. found that land subsidence will cause the loss of soil organic matter, total nitrogen, and available phosphorus, resulting in vegetation degradation [22]. Zhou et al. believed that mining-induced geological deformation is likely to cause irreversible damage to natural groundwater systems and affect the original circulation of groundwater, thus threatening vegetation [23]. Consequently, it is of great significance to study the effects of ground subsidence on vegetation for ecological restoration in mining areas.

However, there is a lack of studies analyzing the effect of ground subsidence on vegetation from multiple scales. Vegetation can be divided into leaf scale and canopy scale. Nowadays, there are more and more studies using remote sensing technology to analyze vegetation degradation [11,24]. The data obtained by remote sensing are the sum of multiple objects on a certain spatial (a pixel). This is the spectral information of a mixed-ground object. Therefore, remote sensing data reflect the information of plants at the canopy scale [25]. However, plant biophysical and biochemical parameters measured in the field by leaf clips reflect leaf scale information [26]. Many studies have focused only on leaf scale or canopy scale [27,28,29,30]. However, the conclusions on different scales may be different. Therefore, the effects of ground subsidence on vegetation should be studied at the leaf scale and canopy scale, respectively, and the similarities and differences between them should be analyzed.

In this manuscript, we used chlorophyll content as an indicator of vegetation degradation. Based on long-term field survey data, we analyzed the effects of ground subsidence on chlorophyll content from two scales (leaf scale and canopy scale) and summarized the similarities and differences. The purpose of this article is to substantiate the theoretical and methodological principles for identifying the impact of soil subsidence on the content of chlorophyll in vegetation in semi-arid mining areas and to develop a methodology for the comprehensive assessment of vegetation degradation in mining areas based on mutil-scale indicators.

## 2. Materials and Methods

### 2.1. Study Area

This experiment took the 31,115 working faces of the Lijiahao coalfield as the research area. The Lijiahao coalfield is located in Ordos, Inner Mongolia Autonomous Region, China. The study area belongs to a semi-arid continental climate zone. The annual average precipitation is 348.3 mm. The annual average evaporation is 2506.3 mm, which is 7.2 times the precipitation. Therefore, this area is short of water resources, and the ecological environment is fragile. The main vegetation types in the study area are *Stipa baicalensis*, *Caragana korshins*, and other drought-tolerant species. The main soil type is loess. The working face parameters and mining parameters are shown in Table 1. The mining method is underground mining, so it will cause a large area of subsidence. During the survey period, this working face was mined for the first time.

### 2.2. Data Acquisition

#### 2.2.1. The Quadrats

We set up three transects on the subsidence area. Each transect included eight quadrats. According to the characteristics of ground subsidence, the subsidence area was divided into three mining disturbance areas: neutral area, compression area, and extension area [31,32]. There were two quadrats each in the neutral area, compression area, extension area, and natural area. The layout of the quadrats is shown in Figure 1. The quadrats in the natural area were far away from the subsidence area. Subsidence areas (neutral area, compression area, and extension area) had the same environmental conditions (soil type, vegetation type, etc.) as natural areas, except that they were affected by mining. The investigation objects and the investigation frequency are shown in Table 2.

#### 2.2.2. Leaf Chlorophyll Content

*Stipa baicalensis* (Sb) and *Caragana korshins* (Ck) were taken as the research objects. They were the main species in the study area. A portable instrument (SPAD-502PLUS, Konica Minolta, Tokyo, Japan) was used to measure the SPAD (Soil and plant analyzer development) value of the leaf, which represented the chlorophyll content at the leaf scale [33,34]. SPAD-502 determined the leaf chlorophyll content by measuring leaf transmittance in two wavelength ranges [35]. Due to the leaf clip, the measurement area was only 2mm x 3mm. Therefore, the measurement results were only related to the biochemical parameters of the tested leaves. There was a stable quantitative relationship between the SPAD value and leaf chlorophyll content [36,37]. For Sb or Ck, the SPAD values of different branches differed. Even for the same branch, the SPAD values in different leaves differed [38]. To reduce the error caused by heterogeneity, the same leaves were measured each time. We selected 10 leaves from each plant and measured each leaf five times. The average of all measurements was calculated.

#### 2.2.3. Ground Subsidence Monitoring

We established 38 fixed monitoring points in the subsidence area to monitor ground subsidence. The distribution of monitoring points is shown in Figure 1. There were 21 monitoring points along the mining direction and 17 monitoring points perpendicular to the mining direction. The monitoring point was made of concrete. The measuring instrument was RTK (the spatial accuracy is ±2cm; TɅRGET, Guangzhou, China). The elevation of the monitoring point was recorded for each measurement. In transect 1, within each quadrat, there is a fixed monitoring point. Subsidence and chlorophyll content were measured simultaneously.

#### 2.2.4. UAV Images

Images were acquired on cloudless days. The unmanned aerial vehicle (UAV) model is PHANTOM4 RTK (DJI, Shenzhen, China). According to the terrain conditions, mining progress, and accuracy requirements, we designed a UAV flight plan, including flight range, flight height, and image overlap. The flight range was the entire subsidence area. The flight altitude was 60m. Image overlap was 80%. Data were collected between 11:00 and 13:00 (i.e., solar noon) to minimize disturbances from the atmosphere and changes in solar elevation. In addition, we set up 20 control points in this area for geometric correction. A total of 11 sets of images were acquired.

### 2.3. Data Processing

#### 2.3.1. UAV Image Processing

UAV images were processed using the professional photogrammetry software Pix4Dmapper. Based on the information of the control points, high-precision orthophotos and point cloud data were generated. Because there were non-ground points (houses, trees, etc.) and noise points in the point cloud, the ground points were separated from them. Therefore, the point cloud data were imported into TerraSolid software for point cloud filtering. In the process of separating ground points, voids were created. Therefore, the Kriging Interpolation Algorithm was used when generating DEM (Digital Elevation Model). This step was performed in ArcGIS 10.6. Finally, the orthophoto and DEM were obtained. The DEM and orthophoto both had a spatial resolution of 2cm.

#### 2.3.2. Canopy Chlorophyll Content

In this study, images were obtained from the UAV platform and contained a lot of information about the plant canopy. Therefore, the remote sensing vegetation index was used to characterize canopy chlorophyll content [21]. Since the images included red band, green band, and blue band, we chose the RGB vegetation index. Previous studies have shown that the red band and blue band are correlated well with chlorophyll content [39]. Therefore, some vegetation indices based on red band and blue band were proposed to indicate chlorophyll content, such as I_kaw_ and BRRI [40,41]. After a comprehensive comparison, we finally chose the BRRI (Blue-red ratio index). Numerous studies have shown that BRRI is a good proxy for chlorophyll content [41,42,43,44]. In particular, Qu et al. also obtained images from PHANTOM4 RTK (the same UAV used in this study) [42]. The formula for BRRI is as follows (Formula 1). B represents the reflectance of the blue band, and R represents the reflectance of the red band. The canopy of each Sb or Ck was extracted from the orthophoto using hand delineation. The non-leaf parts and very bright and shadow pixels within the canopy were masked using a threshold method. The RGB values for the remaining pixels were averaged as the RGB values of the canopy. Subsequently, the BRRI was calculated using RGB values. Chlorophyll content at leaf scale and canopy scale was obtained from the same individual plant.
(1)$$\mathrm{BRRI}=\frac{B}{R}$$

#### 2.3.3. Vegetation Degradation Coefficient

To objectively indicate the degree of vegetation degradation, we constructed a degradation coefficient. The calculation method is shown in Formulas (2) and (3). Comparing the vegetation in the subsidence area with the vegetation in the natural area clearly reflected whether the vegetation was degraded. When the vegetation was not disturbed by mining, the coefficient was around 1 due to similar environmental conditions in the subsidence area and the natural area. However, when the vegetation was disturbed by mining, the degradation coefficient gradually decreased. The smaller the value, the greater the impact of mining on vegetation.
(2)$$\mathrm{degradation}\;\mathrm{coefficient}\;\left(\mathrm{leaf}\;\mathrm{scale}\right)=\frac{\mathrm{SPAD}\left(\mathrm{subsidence}\;\mathrm{area}\right)}{\mathrm{SPAD}\left(\mathrm{natural}\;\mathrm{area}\right)}$$
(3)$$\mathrm{degradation}\;\mathrm{coefficient}\;\left(\mathrm{canopy}\;\mathrm{scale}\right)=\frac{\mathrm{BRRI}\left(\mathrm{subsidence}\;\mathrm{area}\right)}{\mathrm{BRRI}\left(\mathrm{natural}\;\mathrm{area}\right)}$$

#### 2.3.4. Subsidence Parameters

We selected cumulative subsidence (m) and subsidence rate (m/d) as subsidence parameters. Cumulative subsidence referred to the total subsidence compared to before mining. The subsidence rate referred to the subsidence that occurred just the day before. For example, we measured chlorophyll content on 1 July. Cumulative subsidence referred to the total subsidence from before mining to 1 July. The subsidence rate was the cumulative subsidence on 1 July minus the cumulative subsidence on 30 June.

## 3. Results

### 3.1. Time Series Monitoring Results of Ground Subsidence

#### 3.1.1. The Fixed Monitoring Points

Figure 2 shows the ground subsidence data obtained from the fixed monitoring points. The two quadrats in the neutral area were located at points 1 and 2. The two quadrats in the compression area were located at points 6 and 8, and the two quadrats in the extension area were located at points 10 and 12. Point 17 was in the natural area. The subsidence from large to small was: neutral area>compression area>extension area, and the No. 17 monitoring point basically had no subsidence. According to the underground mining parameters, the working face was mined to the position of transect 1 on 15 June. Due to the advanced influence, transect 1 slightly sank on 1 June, with a maximum subsidence value of 0.074 m (point 2). When the mining range exceeded transect 1, the monitoring points continued to sink and were in the active period of sinking. After 8 July, the ground gradually entered a stable state. During the investigation, the maximum subsidence was 4.338 m.

#### 3.1.2. The DEM Generated from UAV Images

For transect 1, we used fixed monitoring points to obtain subsidence data. For transects 2 and 3, we obtained subsidence data in the form of drone aerial photography. Figure 3 was obtained by subtracting the DEM of 1 September from the DEM of 30 May. Surface subsidence caused by mining created a basin in the study area. According to Figure 3, during the investigation period, the maximum subsidence value in this area was 4.32 m, which was very close to the 4.338 m obtained through the field measurement. Comparing the elevation value obtained by the DEM with the elevation value obtained by the field measurement, the difference was small (Figure 4). Therefore, the DEM obtained by the UAV monitored ground subsidence.

### 3.2. Time Series Monitoring Results of Chlorophyll Content

#### 3.2.1. The Leaf Scale

Since the plants were in the growing season, the chlorophyll content in the natural area showed an increasing trend during the investigation period. In general, the chlorophyll content in the subsidence area also showed an increasing trend, but the increase rate was obviously slower than that in the natural area. The difference in chlorophyll content between the subsidence area and the natural area first increased and then decreased. In some periods, chlorophyll content even declined in the subsidence area, which means that the plants were severely disturbed.

The degradation coefficient reflected the difference in chlorophyll content between the subsidence area and the natural area. It can be seen from Figure 5 that the degradation coefficients of Sb and Ck decreased first and then increased, and the turning point was 8 July. At the beginning of the investigation, the degradation coefficients were all around 1, and the degradation coefficients of most of the quadrats were even greater than 1. This indicates that the original state of plants in the subsidence area and the natural area is similar, and even the growth state of the plants in the subsidence area is better than that in the natural area.

As the ground sank, the degradation coefficient decreased gradually, indicating that the chlorophyll content of plants in the subsidence area was obviously lower than that in the natural area. Studies have shown that underground mining will result in surface subsidence and the formation of fissures, which will lead to the decline of soil water content, the deterioration of soil physical and chemical properties, and even the rupture of plant roots. These will have a non-negligible negative impact on vegetation growth. According to the minimum value of the degradation coefficient (on 8 July), it can be found that whether it was Sb or Ck, the value in the neutral area was the largest, followed by the compression area and the extension area. This showed that coal mining had the greatest impact on the plants in the extension area, followed by the compression area, and the least impact on the plants in the neutral area. Comparing the degradation coefficients of Sb and Ck, it can be seen that under the same ground subsidence, the degradation coefficient of Ck was greater than that of Sb, which indicates that Ck has better resistance to mining.

After 8 July, the degradation coefficient showed an increasing trend, indicating that the plants in the subsidence area grew rapidly, and the chlorophyll content increased significantly. It can be seen from Figure 2 that after 8 July, the land subsidence rate decreased significantly, and ground activities basically stopped. Therefore, the impact of ground subsidence on plant growth was greatly reduced. On the other hand, in order to restore the ecological environment of the subsidence area and promote the recovery of damaged plants, the managers began to water the plants in the subsidence area and filled a large number of fissures after 10 July. Therefore, the plants in the subsidence area recovered gradually and grew rapidly. Although managers have taken some measures to promote the recovery of the plants, until the end of the investigation, the degradation coefficient was still less than 1, which means that the growth status of the plants in the subsidence area was still poor. Therefore, the ecological restoration of mining areas is a long-term project that requires continuous and scientific effort from managers.

#### 3.2.2. The Canopy Scale

In Figure 6, the trend of the degradation coefficient is similar to that in Figure 5. Consistent with the results of the leaf scale, land subsidence had the greatest impact on plants in the extension area, followed by the compression area, and the impact on plants in the neutral area was the least. In addition, in the same subsidence, the degradation coefficient of Ck was greater than that of Sb, indicating that coal mining had less impact on Ck. After 3 July, the degradation coefficient gradually increased, which was consistent with the results of the leaf scale. This was mainly because there was no longer severe subsidence, and managers had taken timely ecological restoration measures.

### 3.3. The Correlation between Degradation Coefficient and Subsidence Parameters

We analyzed the correlation between subsidence parameters and degradation coefficient at the leaf and canopy scales (Table 3 and Table 4). Data were analyzed using the SPSS 22.0 statistical package. A bivariate correlation procedure was used to calculate the Pearson correlation coefficient. Both the leaf scale and canopy scale results showed that there was no correlation between the degradation coefficient and subsidence rate, while the correlation between the degradation coefficient and cumulative subsidence was strong. In this study, the subsidence rate was equal to the subsidence value the previous day. This demonstrates that ground subsidence will not immediately affect the growth of vegetation. It is the cumulative subsidence that threatens vegetation growth. The effect of ground subsidence on plants characterized the hysteresis effect and cumulative effect. Therefore, if some management measures are taken in time, the impact of mining may be mitigated. However, if managers do not take any action, ground deformation will severely affect vegetation growth.

### 3.4. The Relationship between Degradation Coefficient and Cumulative Subsidence

According to Table 3 and Table 4, the degradation coefficient and cumulative subsidence have a stronger correlation than the subsidence rate. Hence, we drew scatter diagrams between the degradation coefficient and cumulative subsidence at the leaf and canopy scales, respectively (Figure 7 and Figure 8). Managers took some ecological recovery measures after 10 July, and the surface subsidence basically stopped. Hence, we only used data before 10 July. According to the fitting results, the slope in the neutral area was the largest, followed by the compression area, and the slope in the extension area was the smallest. This showed that subsidence had the least impact on plants in the neutral area and had the greatest impact on plants in the extension area. This was consistent with the results obtained in Section 3.2. In addition, the slope at the canopy scale was smaller than the leaf scale for the same species in the same mining disturbance area. That is, at the canopy scale, vegetation degradation was more severe than at the leaf scale. Therefore, when we analyze the impact of subsidence on vegetation from different scales, the results may differ. We should be concerned about the differences in the results due to the scale.

## 4. Discussion

### 4.1. The Degradation Coefficient

Instead of directly using SPAD or BRRI to analyze vegetation degradation, we chose the degradation coefficient. Because the measured SPAD or remote sensing generated BRRI cannot directly reflect the vegetation degradation degree. However, the degradation coefficient compares the vegetation in the subsidence area with the vegetation in the natural area, so it can clearly reflect whether the vegetation in the subsidence area is degraded. For example, in Figure 5, the smallest degradation coefficient is 0.774, indicating that the chlorophyll content in the subsidence area is only 77.4% of that in the natural area. This means that the vegetation has been severely damaged.

### 4.2. The Similarities between the Leaf Scale and the Canopy Scale

The results of both leaf scale and canopy scale showed that the degradation coefficient in neutral area is the largest, followed by compression area and extension area. This indicates that coal mining has the least effect on plants in the neutral area, followed by the compression area, and the greatest effect on plants in the extension area. It can be seen from Figure 2 that the subsidence value in the neutral area is the largest because the neutral area is located in the center of the subsidence basin. The extension area has the least subsidence because it is located at the edge of the basin. However, coal mining will not only cause vertical movement of the ground but also horizontal movement [45,46]. Although the subsidence in the neutral area is the largest, it is easier to form fissures in the extension area due to the land being stretched. Fissures will greatly reduce soil moisture [47]. In particular, the study area is located in a semi-arid region, and soil moisture is crucial for plant growth. Therefore, plant degradation in the extension area is more serious. Although there was obvious ground subsidence in the neutral area, the fissures were less, so the plants were slightly disturbed. Consequently, the results of both the leaf scale and canopy scale indicate that the extension area is the key area for ecological restoration in the mining area. Timely filling ground fissures is a wise choice.

Figure 5 (leaf scale) and Figure 6 (canopy scale) all showed that subsidence had a slight effect on the chlorophyll content of Ck but a serious effect on the chlorophyll content of Sb. This indicates that Ck is more suitable as a pioneer species of ecological restoration in this area. We think this is because Ck has stronger roots. On the one hand, the roots of Sb are more likely to break under severe ground deformation (vertical and horizontal movement), while Ck can survive (Figure 9). On the other hand, drought is the biggest threat to plants in this region. The stronger roots could help Ck gain access to more groundwater resources and thus resist drought. After comparing various plants in the Wuhai coalfield (also located in the Inner Mongolia Autonomous Region), Tan also pointed out that Ck had a strong ability to enrich soil water and nutrients, and was suitable for ecological restoration in the mining area [48].

At both the leaf scale and canopy scale, there is no correlation between subsidence rate and degradation coefficient, but there is a significant correlation between cumulative subsidence and degradation coefficient. In this manuscript, the subsidence rate is equal to the subsidence value the previous day. This shows that ground subsidence does not immediately disturb the growth of vegetation. Of course, there are extremes. Because the ground sinks at inconsistent rates, subsidence induces fissures (Figure 10). Fissures may break the roots of a plant, causing it to die in a short time. However, if the roots are not destroyed, the effects of subsidence on plant chlorophyll content have a hysteresis effect. This gives us a chance to take remedial action. Therefore, if managers take timely ecological restoration measures at the initial stage of subsidence, more serious vegetation degradation may be avoided. The strong correlation between cumulative subsidence and the degradation coefficient indicated that the effects of subsidence on plants have a cumulative effect. This also warns us that ecological restoration should be carried out in time. Hu et al. put forward the concept and technology of the “Concurrent Mining and Reclamation (CMR)” of the coal mine’s ecological environment and the realization path [49]. It is believed that CMR is based on the concept of “Source and Process Control,” rather than the traditional concept of “Terminal Treatment,” which is characterized by synchronous or timely treatment during the mining process [50]. Chugh pointed out that CMR was an advanced technology in subsidence land reclamation [51].

### 4.3. The Differences between Leaf Scale and Canopy Scale

It can be seen from Figure 7 and Figure 8 that the slope at canopy scale is smaller than that at the leaf scale. This means that the degradation coefficient decreased more rapidly at the canopy scale when the ground subsided. Consequently, when assessing vegetation degradation, the results obtained by remote sensing were more severe than those measured in the field.

We believe that this is because the canopy chlorophyll content obtained by the remote sensing vegetation index is not only related to the leaf chlorophyll content but also affected by the canopy structure. When the canopy structure changes, the quantitative relationship between the canopy chlorophyll content and the leaf chlorophyll content also changes. For example, Gitelson et al. and Fu et al. believed that canopy chlorophyll content was the product of leaf chlorophyll content and the leaf area index [52,53]. Coal mining creates many fissures in the subsidence area. Fissures can make plants more vulnerable to drought stress. Studies have shown that in order to retain more water, the vegetation canopy structure will change under drought conditions [54,55,56]. Therefore, when plants are affected by mining, canopy structure parameters will change, and the quantitative relationship between canopy chlorophyll content and leaf chlorophyll content will also change. We used data from different periods to build the BRRI-SPAD model (Figure 11). The data acquired on 11 June is represented by a black square, and the data acquired on 1 July is represented by a red circle. According to Figure 11, when plants are disturbed by mining, the quantitative model between BRRI and SPAD will change, and the slope of the model will become smaller.

We further illustrate this phenomenon with a simplified diagram (Figure 12). Before mining, no matter at the leaf scale or canopy scale, chlorophyll content was basically the same in the natural area and subsidence area, and the degradation coefficient was about 1. After a period of mining, the chlorophyll content continued to increase in the natural area but gradually decreased in the subsidence area. Therefore, the degradation coefficient at the leaf scale is $\frac{T2}{T1}$. If the quantitative model between leaf chlorophyll content and canopy chlorophyll content does not change, the degradation coefficient should be $\frac{V2}{V1}$ at canopy scale. However, when the canopy structure changes, the leaf chlorophyll content—canopy chlorophyll content model changes from model 1 to model 2. Therefore, the degradation coefficient of the canopy scale changes from $\frac{V2}{V1}$ to $\frac{V3}{V1}$. Obviously,$\;\frac{V3}{V1}$ < $\frac{V2}{V1}$ = $\frac{T2}{T1}$. Consequently, the vegetation degradation obtained by remote sensing is more severe than that measured in the field.

### 4.4. The Shortcomings and Future Work

In this study, PHANTOM4 RTK was used to acquire remote sensing images. This is because PHANTOM4 RTK can acquire both the orthophoto and DEM. In addition, the PHANTOM4 RTK is easy to operate and suitable for repeated experiments. For example, in this study, we obtained 11 sets of images. However, the PHANTOM4 RTK has only three bands (red, green, and blue). Based on the RGB images, we calculated the BRRI index. Although the BRRI index has been shown to correlate well with chlorophyll content in many studies, this experiment should be repeated using other vegetation indices in the future. For example, the vegetation index is based on hyperspectral remote-sensing images. Hyperspectral has more bands, so it can better characterize chlorophyll content [57,58,59].

In addition, vegetation degradation in mining areas is reflected in many aspects. For example, vegetation coverage [60], chlorophyll fluorescence [31], biodiversity [61], etc. In the future, we should study the effects of ground subsidence on these factors at different scales and analyze their similarities and differences. Like chlorophyll content, chlorophyll fluorescence can also be divided into leaf scale and canopy scale [62,63,64]. Therefore, we can study the effect of ground subsidence on chlorophyll fluorescence from the leaf scale and canopy scale. For vegetation cover and biodiversity, their definitions are based on community. However, the size of the community is not fixed. When the defined community size is small, we can obtain vegetation coverage or biodiversity through field surveys, such as quadrats. When the defined community size is large, we can obtain it by remote sensing. However, when the community size is different, the results differ [65,66]. Therefore, we should study the effect of ground subsidence on vegetation coverage or biodiversity from different scales. In conclusion, we should pay attention to the scale effect when evaluating the impact of coal mining or ground subsidence on the ecological environment. Only when we evaluate the multi-scale can we get more comprehensive and objective evaluation results. Our research conclusions can better serve the government. However, sometimes we cannot evaluate the effect of coal mining on the ecological environment from multiple scales when time-cost is taken into account. Therefore, we should choose the most appropriate scale. How to obtain an optimal scale should also be one of our future research directions.

## 5. Conclusions

In this study, we analyzed the effect of ground subsidence on chlorophyll content at different scales. We found that the following three conclusions can be reached at both the leaf scale and canopy scale: (1) coal mining has the least effect on chlorophyll content in the neutral area, followed by the compression area, and the greatest effect on chlorophyll content in the extension area; (2) subsidence had a slight effect on chlorophyll content of *Caragana korshins*, but a serious effect on chlorophyll content of *Stipa baicalensis*; and (3) chlorophyll content is not immediately affected when the ground sinks. The effect of ground subsidence on plants characterizes the hysteresis effect and cumulative effect. It is the cumulative subsidence that affects chlorophyll content. However, we found that the degradation coefficient decreased more rapidly at the canopy scale when the ground subsided. The vegetation degradation at the canopy scale obtained by remote sensing is more severe than that at leaf scale measured in the field. Therefore, when evaluating vegetation degradation, the results obtained at different scales differ. This difference is due to the properties of the observed objects themselves.

Based on the above conclusions, we put forward specific practical recommendations. We pointed out that ground fissures are the key to ecological reclamation. Managers should fill the fissures in time to prevent the loss of soil nutrients and soil moisture, especially in extension areas. In addition, *Caragana korshins* ought to be regarded as the pioneer species in semi-arid mining areas and be widely cultivated. We recommend that mining and ecological restoration be carried out concurrently. We believe that “Concurrent Mining and Reclamation (CMR)” is an advanced technology in subsidence land reclamation and should be widely promoted. Most importantly, in the future, managers should assess plant degradation on multiple scales, such as the leaf scale, canopy scale, and even landscape scale, rather than just a single scale. At leaf scale, we can use field measurement methods, such as a variety of hand-held instruments. At the canopy or landscape scale, we can use remote sensing. However, managers need to be aware of the differences in different scales, and integrate the results of multiple scales to comprehensively evaluate environmental degradation in mining areas. In addition, the degradation coefficient proposed in this study is suitable for indicating vegetation degradation and is recommended to be widely used.

## Figures and Tables

**Figure 1 ijerph-20-00493-f001:**
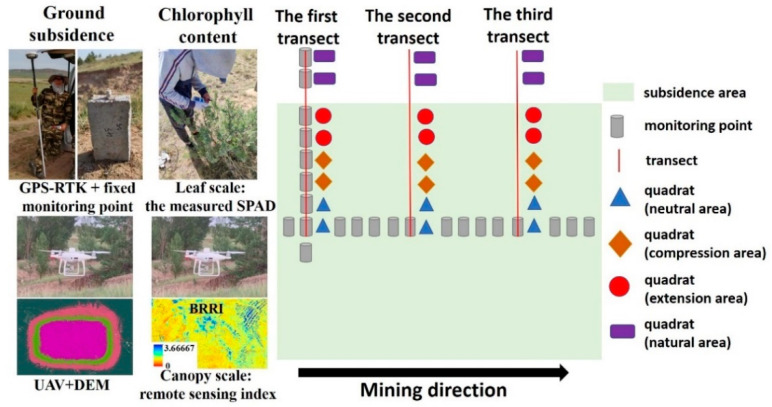
The layout of the transects and quadrats. Source: Built on the basis of the author’s field survey.

**Figure 2 ijerph-20-00493-f002:**
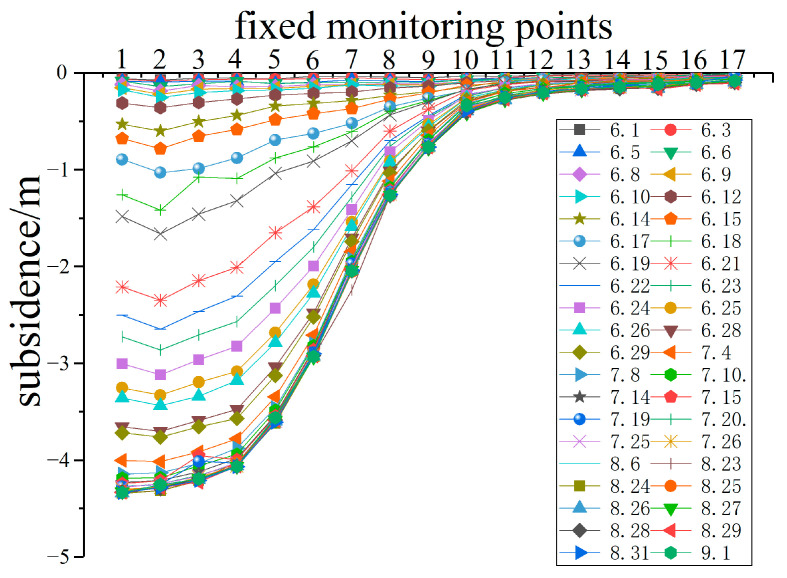
Subsidence data obtained from fixed monitoring points (perpendicular to the mining direction). The legend is the date of the measurement. Source: Built on the basis of the author’s field survey and calculations.

**Figure 3 ijerph-20-00493-f003:**
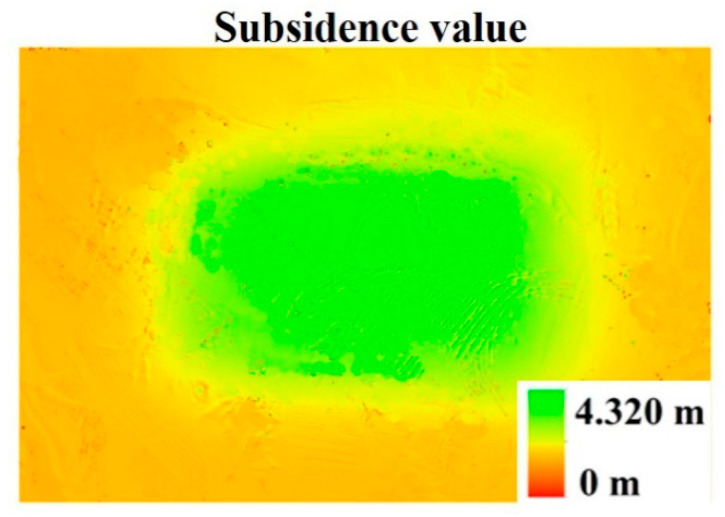
Ground subsidence of the study area. The figure was obtained by subtracting the DEM of 1 September from the DEM of 30 May. Source: Built on the DEM obtained from UAV images.

**Figure 4 ijerph-20-00493-f004:**
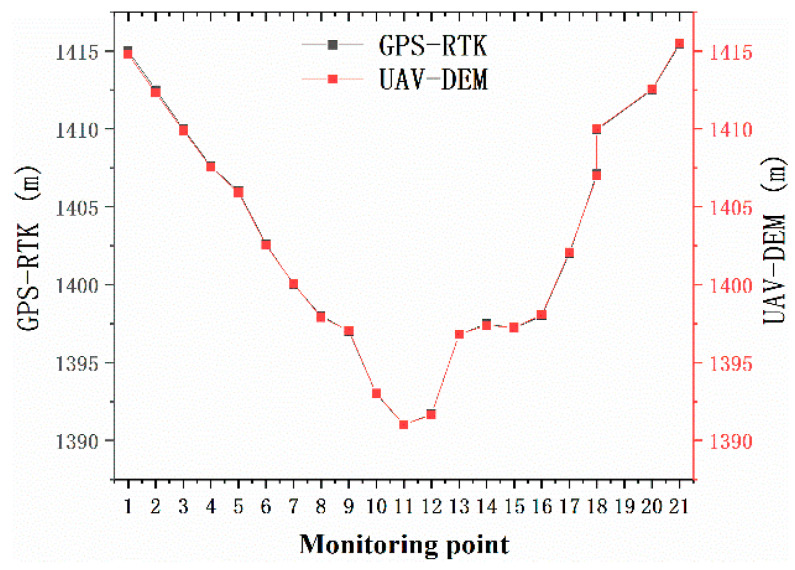
The elevation value obtained by the DEM and the elevation value obtained by the field measurement (along the mining direction). Source: Built on the basis of the author’s field survey and calculations.

**Figure 5 ijerph-20-00493-f005:**
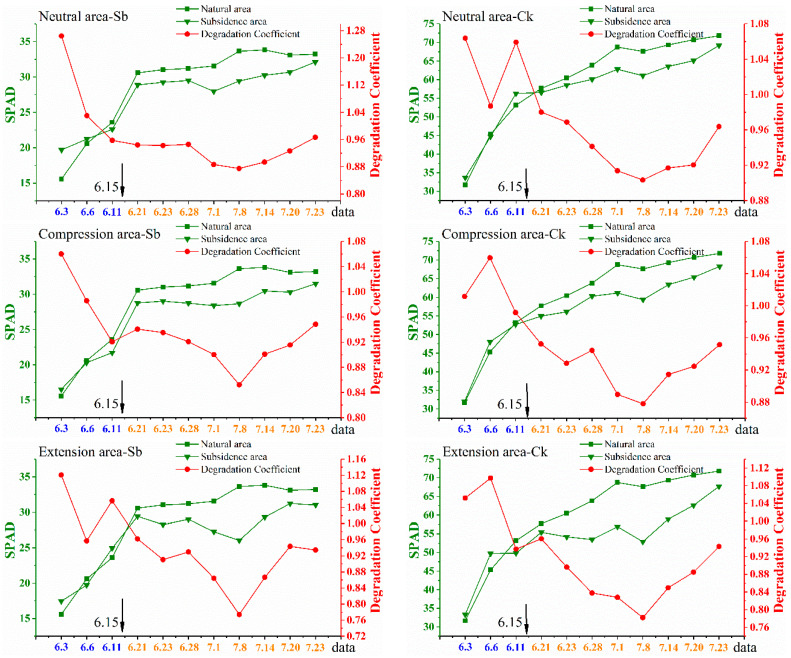
Time series monitoring results of chlorophyll content at leaf scale. All the data in the figures were obtained at transect 1. The 31,115 working faces were mined to the position of transect 1 on 15 June. There were two quadrants in each disturbance area, and only the data from the first quadrat are shown in the figures. Source: Built on the basis of the author’s field survey and calculations.

**Figure 6 ijerph-20-00493-f006:**
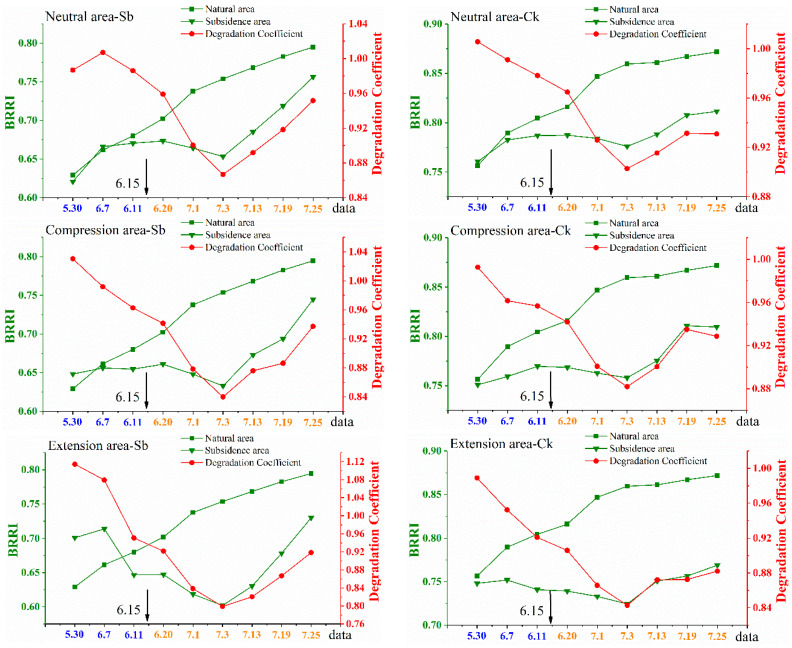
Time series monitoring results of chlorophyll content at canopy scale. All the data in the figures were obtained at transect 1. The 31,115 working faces were mined to the position of transect 1 on 15 June. There were two quadrants in each disturbance area, and only the data from the first quadrat are shown in the figures. Source: Built on the basis of the author’s field survey and calculations.

**Figure 7 ijerph-20-00493-f007:**
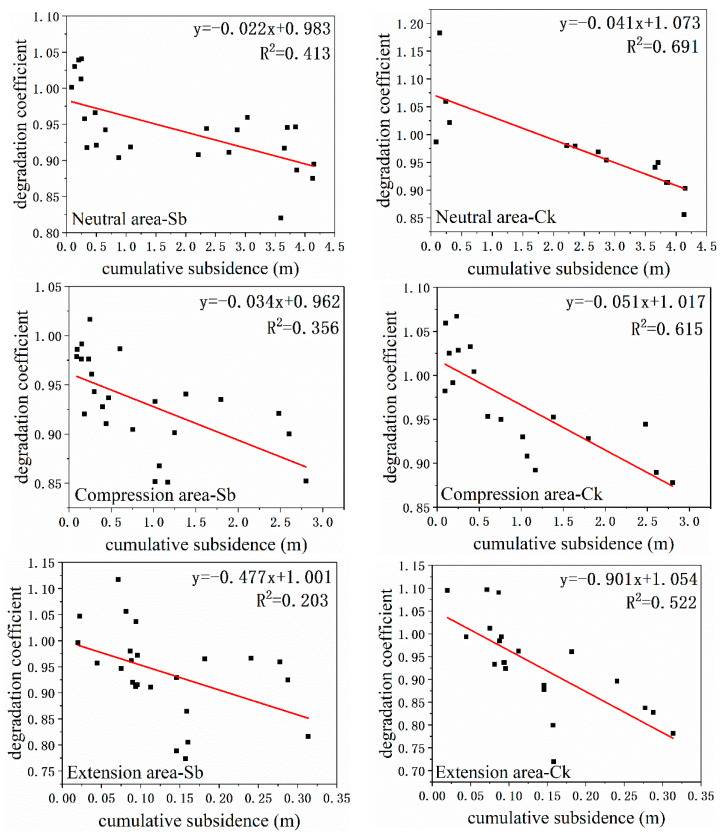
Relationship between degradation coefficient and ground subsidence at leaf scale. Source: Built on the basis of the author’s calculations.

**Figure 8 ijerph-20-00493-f008:**
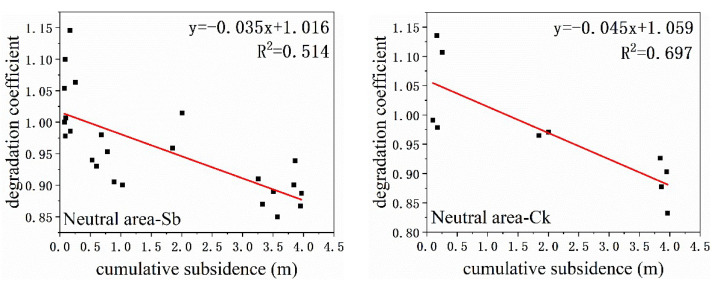
Relationship between degradation coefficient and ground subsidence at canopy scale. Source: Built on the basis of the author’s calculations.

**Figure 9 ijerph-20-00493-f009:**
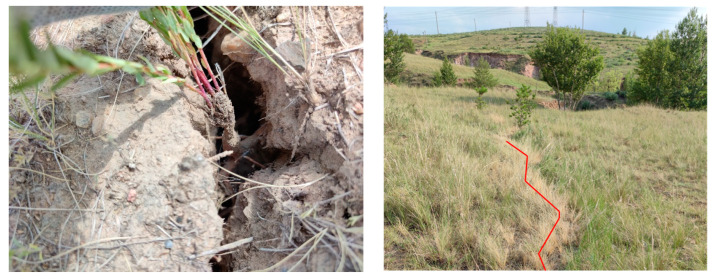
Even if there are fissures, the roots of Ck will not easily break. Ck can continue to live (**left** figure). However, Sb is more likely to die (there are fissures where the red line is marked on the figure) (**right** figure). Source: Built on the basis of the author’s field survey.

**Figure 10 ijerph-20-00493-f010:**
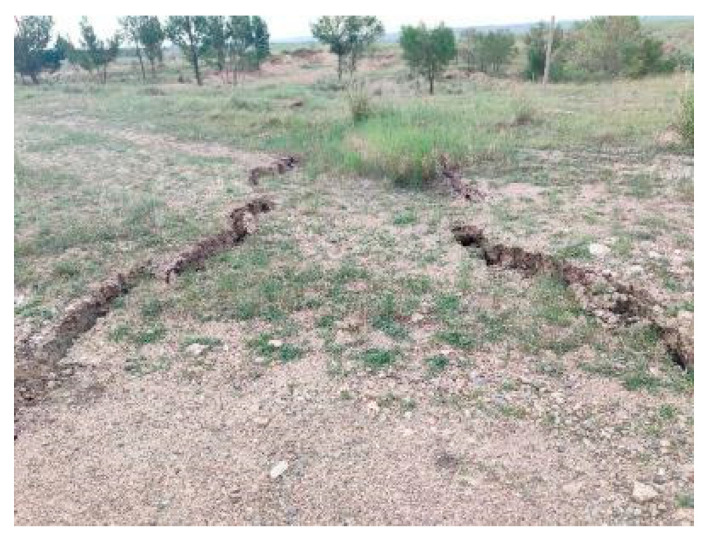
Fissures will form when the ground sinks at inconsistent rates. Plants located directly above the fissures will die immediately, but those around the fissures will still survive. Source: Built on the basis of the author’s field survey.

**Figure 11 ijerph-20-00493-f011:**
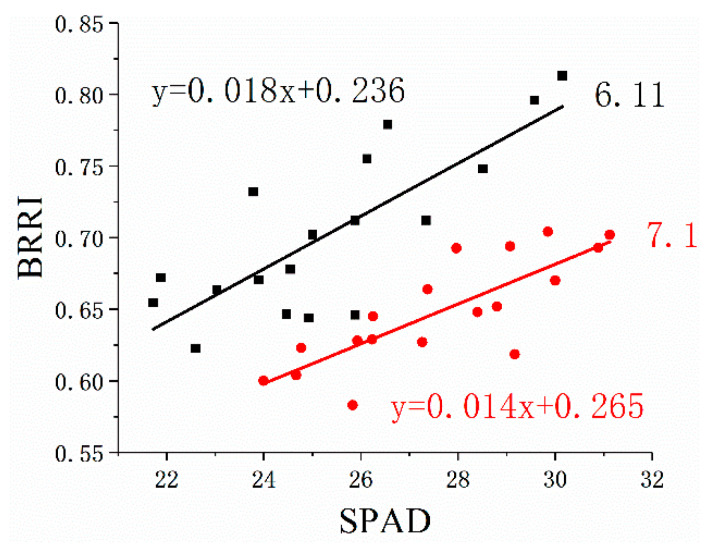
BRRI-SPAD model for different periods. The data acquired on 11 June is represented by a black square, and the data acquired on 1 July is represented by a red circle. The data were obtained at 11:00–13:00, and the weather conditions were the same. Source: Built on the basis of the author’s field survey and calculations.

**Figure 12 ijerph-20-00493-f012:**
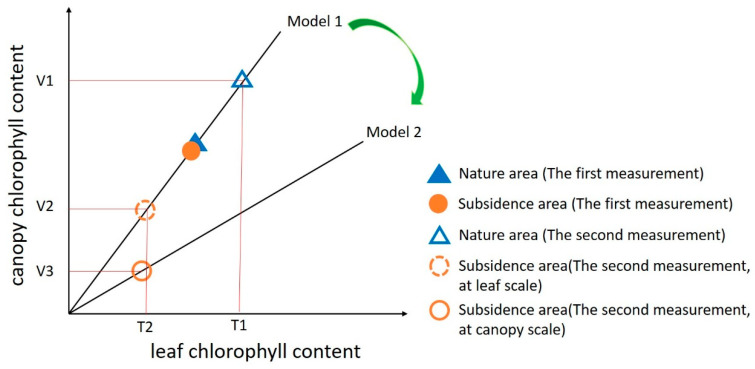
The difference in degradation coefficient at canopy scale and leaf scale. Before mining, no matter at the leaf scale or canopy scale, chlorophyll content was basically the same in the natural area and subsidence area, and the degradation coefficient was about 1. After a period of mining, the chlorophyll content continued to increase in the natural area but gradually decreased in the subsidence area. Therefore, the degradation coefficient at the leaf scale is $\frac{T2}{T1}$. If the quantitative model between leaf chlorophyll content and canopy chlorophyll content does not change, the degradation coefficient should be $\frac{V2}{V1}$ at canopy scale. However, when the canopy structure changes, the model changes from model 1 to model 2. Therefore, the degradation coefficient at the canopy scale changes from $\frac{V2}{V1}$ to $\frac{V3}{V1}$. Obviously, $\;\frac{V3}{V1}$ < $\frac{V2}{V1}$ = $\frac{T2}{T1}$. Source: Author’s development.

**Table 1 ijerph-20-00493-t001:** Working face parameters and mining parameters.

Parameters	Value
length of working face	2600 m
width of working face	600 m
coal seam dip angle	0–3°
mining depth	122–266 m
mining height	4.7–4.9 m
mining rate	8 m/d

Source: These parameters were provided by the managers of the Lijiahao coalfield.

**Table 2 ijerph-20-00493-t002:** Investigation objects and investigation frequency.

Investigation Content	Investigation Frequency
Chlorophyll content at leaf scale	Transect 1: a total of 11 sets of measurements.
	Transect 2: a total of 11 sets of measurements.
	Transect 3: a total of 11 sets of measurements.
Ground subsidence	a total of 40 sets of measurements.
UAV images	a total of 11 sets of images.

Source: Built on the basis of the author’s field survey.

**Table 3 ijerph-20-00493-t003:** The Pearson correlation coefficient between the degradation coefficient and subsidence parameters at leaf scale.

	Cumulative Subsidence	Subsidence Rate
Neutral area-Sb	−0.642 **	−0.298
Compression area-Sb	−0.597 **	−0.244
Extension area-Sb	−0.451 *	−0.148
Neutral area-Ck	−0.831 **	−0.354
Compression area-Ck	−0.784 **	−0.262
Extension area-Ck	−0.723 **	−0.211

Note: ** means that the correlation coefficient passes the 0.01 significance test. * means that the correlation coefficient passes the 0.05 significance test. Source: Built on the basis of the author’s calculations.

**Table 4 ijerph-20-00493-t004:** The Pearson correlation coefficient between the degradation coefficient and subsidence parameters at the canopy scale.

	Cumulative Subsidence	Subsidence Rate
Neutral area-Sb	−0.717 **	−0.312
Compression area-Sb	−0.651 **	−0.267
Extension area-Sb	−0.629 **	−0.178
Neutral area-Ck	−0.835 **	−0.332
Compression area-Ck	−0.619 *	−0.234
Extension area-Ck	−0.674 **	−0.219

Note: ** means that the correlation coefficient passes the 0.01 significance test. * means that the correlation coefficient passes the 0.05 significance test. Source: Built on the basis of the author’s calculations.

## Data Availability

Data are not publicly available, though the data may be made available on request from the corresponding author.

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
