# Peer review of "Effects of Ground Subsidence on Vegetation Chlorophyll Content in Semi-Arid Mining Area: From Leaf Scale to Canopy Scale"

_ijerph, 2022, doi:10.3390/ijerph20010493_

Round 1

Reviewer 1 Report

minor revision

Q1:

1. Introduction

“The coal resources in the west of China accounts for more than 70% of the total coal resources. Unfortunately, the west is a semi-arid region with extremely fragile ecological environment.” Please point out the source of the 70%. Is western China all semi-arid region? Relevant references need to be added.

Q2:

2.2.2 Leaf chlorophyll content

“The vegetation degradation caused by coal mining is reflected in many aspects, such as vegetation coverage [27], biomass [28], and chlorophyll content [29]. As an important vegetation biochemical component, chlorophyll content can be used to indicate vegetation degradation [30]. ” This part repeats the second paragraph of the introduction and should be placed in the introduction. Stipa baicalensis and Caragana korshins should be mentioned at the beginning of this paragraph.

Q3:

2.2.2 Leaf chlorophyll content

“To reduce the error caused by heterogeneity, the same leaves were measured each time.” How do you make sure you measure the same leaves every time?

Q4:

3.1.1 The fixed monitoring point

“The two quadrats in the neutral area were located at points 1 and 2. The two quadrats in the compression area were located at points 6 and 8 and the two quadrats in the extension area were located at points 10 and 12.” Why are points 1 and 2 in the neutral area, points 6 and 8 in the compression area, and points 10 and 12 in the extension area? How are the neutral area, compression area and extension area defined?

Q5:

4.4 The shortcomings and future work

“For example, vegetation coverage [58], chlorophyll fluorescence [29], biodiversity [59], etc. In the future, we should study the effects of ground subsidence on these factors at different scales and analyze the similarities and differences.” Why is it possible to study vegetation coverage, chlorophyll fluorescence and biodiversity at different scales? What scales do they have? For example, chlorophyll content can be divided into leaf scale and canopy scale.

Round 2
